# Prognostic Factors in Advanced Non-Small Cell Lung Cancer Patients Treated with Immunotherapy

**DOI:** 10.3390/cancers15194684

**Published:** 2023-09-22

**Authors:** Danilo Rocco, Luigi Della Gravara, Angela Ragone, Luigi Sapio, Silvio Naviglio, Cesare Gridelli

**Affiliations:** 1Department of Pulmonary Oncology, AORN dei Colli Monaldi, 80131 Naples, Italy; danilo.rocco@ospedalideicolli.it; 2Department of Precision Medicine, Università degli Studi della Campania “Luigi Vanvitelli”, 80138 Naples, Italy; luigi.dellagravara@studenti.unicampania.it (L.D.G.); luigi.sapio@unicampania.it (L.S.); silvio.naviglio@unicampania.it (S.N.); 3Max-Planck Institute of Molecular Physiology, 44227 Dortmund, Germany; angela.ragone@unicampania.it; 4Division of Medical Oncology, “S.G. Moscati” Hospital, Contrada Amoretta, 83100 Avellino, Italy

**Keywords:** NSCLC, immunotherapy, prognostic factors, PD-L1, TMB, NLR, TILs, gut microbiota, concomitant medications

## Abstract

**Simple Summary:**

Immunotherapy in the form of Immune Checkpoint Inhibitors which currently represent one of the staples of advanced Non-Small Cell Lung Cancer treatment. However, as of today, no prognostic factors are taken in account to shape immunotherapy-treated patient selection and management according to the most authoritative international guidelines. Therefore, the identification and evaluation of possible prognostic factors in this subset of patients represent an extremely interesting and relevant topic.

**Abstract:**

Taking into account the huge epidemiologic impact of lung cancer (in 2020, lung cancer accounted for 2,206,771 of the cases and for 1,796,144 of the cancer-related deaths, representing the second most common cancer in female patients, the most common cancer in male patients, and the second most common cancer in male and female patients) and the current lack of recommendations in terms of prognostic factors for patients selection and management, this article aims to provide an overview of the current landscape in terms of currently available immunotherapy treatments and the most promising assessed prognostic biomarkers, highlighting the current state-of-the-art and hinting at future challenges.

## 1. Advanced NSCLC: Epidemiology and Subtypes

The Global Cancer Observatory recorded 19.3 million tumor cases worldwide, as well as 9.9 million tumor-related deaths in 2020. In particular, lung cancer accounts for 2,206,771 of the cases and for 1,796,144 of the cancer-related deaths, representing the second most common cancer in female patients (after breast cancer) and the most common cancer in male patients; in addition, it represents the major cause of death in male patients while being the second most common cause of cancer death in female patients (after breast cancer) [1]. Similarly, according to the American Cancer Society projections for 2022, 1,918,030 tumor cases have occurred in the USA, alongside 609,360 tumor-related deaths. In particular, lung cancer has been responsible for 236,740 of the cases and for 130,180 of the cancer-related deaths, representing the second most common tumor in female patients (after breast cancer), the second most common tumor in male and female patients (after breast cancer), and the second most common tumor in male patients (after prostate cancer); moreover, it has represented the major cause of death in male patients, in females, and in males and females [2,3]. With respect to the histological definition, lung cancer can be classified in two major subtypes: Non-Small Cell Lung Cancer (NSCLC, 85% of all the reported lung cancer cases) and Small Cell Lung Cancer (SCLC, 15% of all the reported lung cancer cases) [4,5,6]. In turn, NSCLC itself is further divided into three subgroups: adenocarcinoma (40–50% of all the reported NSCLC cases), squamous cell carcinoma (20–30% of all the reported NSCLC cases), and large cell carcinoma/not otherwise specified (10–20% of all the reported NSCLC cases) [7,8,9]. With reference to the stage at diagnosis, approximately 50–60% of patients are diagnosed with an advanced disease (stage IV of the TNM classification), approximately 20–25% of patients are diagnosed with a locally advanced disease (stage III of the TNM classification), and approximately 20–25% of patients are diagnosed with an early disease (stage I/II of the TNM classification) [10,11,12].

## 2. Immunotherapy in the Form of ICIs: Mechanisms of Action

All of the current FDA (U.S. Food and Drug Administration) and/or EMA (European Medicines Agency)-approved and ASCO (American Society of Clinical Oncology) and/or ESMO (European Society for Medical Oncology)-recommended immunotherapeutic drugs for the treatment of advanced NSCLC fall under the category of immune checkpoint inhibitors (ICIs) and are represented by: nivolumab, pembrolizumab, cemiplimab (anti PD-1 mAbs); atezolizumab, and durvalumab (anti PD-L1 mAbs); and ipilimumab and tremelimumab (anti CTLA-4, i.e., Cytotoxic T-Lymphocyte Antigen 4 mAbs). Immune checkpoints (ICs) can be described as receptor–ligand pairs that, upon binding, stimulate (positive ICs) or inhibit (negative ICs) immune system activity—especially cytotoxic T-cells’ activity In a physiological setting, ICs help in maintaining and regulating the immunological tolerance and in fine-tuning the immune system responses. However, cancer cells can exploit negative ICs to escape immune surveillance. As of today, the two best studied and understood negative immune checkpoints are represented by PD-1-PD-L1 and by CTLA-4-B7. The PD-L1 (Programmed Death Ligand 1) protein can be found on the surface of the antigen-presenting cells (APCs) that inhibit T-cells’ activity upon ligation with its receptor PD-1 (Programmed Death Protein 1), which is located on the T-cells’ surface. In this vein, tumor cells can exploit this IC, expressing PD-L1 on their surface. This binding takes place primarily in peripheral tissues. Similarly, the B7.1/2 protein can be found on the surface of APCs, while its receptor CTLA-4 can be located on the surface of T cells; upon ligation, this receptor–ligand pair blocks T-cell activity. This binding takes place primarily in lymph nodes. Similarly to PD-L1, tumor cells can also express CTLA-4 on their surface. In summary, ICIs exert their activity by stopping these receptor–ligand bindings, preventing cancer cells from exploiting these negative ICs, and thus, inhibiting the cytotoxic T-cells’ activity, in turn in turn the immune surveillance and promoting cytotoxic T-cell-mediated cancer cell death [13,14,15,16,17,18,19,20,21,22,23,24,25].

Apart from the established role of the immune checkpoints targeted by currently approved and recommended ICIs, it is worth mentioning that the role of several other ICs is currently being investigated in order to better understand the interplay between ICs in the wider context of the tumor microenvironment and to find new suitable targets. However, as of today, these new targets are still assessed at an investigational level. The T-cell immunoglobulin and mucin-domain containing-3 (TIM-3), lymphocyte activation gene-3 (LAG-3), V-domain Ig suppressor of T-cell activation (VISTA), and OX40 and glucocorticoid-induced TNFR-related protein (GITR) are part of the most promising new ICs; with the exception of OX40 and GITR, they all act as negative ICs members [26,27]. TIM-3 is expressed on the surface of different kinds of T-cells (CD4+ cells, CD8+ cells [28], T-reg cells [29], and Th17 cells) [30], but also on the surface of APCs, and it exerts its activity upon binding the galectin-9 expressed on tumor cells. It is believed to play a role in the development of immune tolerance, and it also seems linked to T-cells’ exhaustion. In fact, its upregulation on CD8+ T cells reduces their activity and [31,32] its upregulation on T reg cells [33] enhances their performance [34]. In addition, TIM-3 is also expressed on Tumor-Infiltrating Lymphocytes and the higher levels of expression seem to be associated with a negative prognostic outcome [35]. The preclinical data point to the fact that a high TIM-3 expression is linked to resistance to anti PD-1 ICIs, thus highlighting a possible role for a dual PD-1/TIM-3 blockade treatment strategy [36,37]. Similarly to TIM-3, LAG3 is also expressed on the surface of T cells (especially TILs) and is also part of another negative IC exerting its activity upon binding its binding partners: the class II MHC, galectin-3, and LSECtin, which are expressed on the surface of tumor cells [38]. While in a physiological setting, the LAG3 pathway is activated in order to regulate T-cell activation [39], and to curb inflammatory responses [40], this pathway can also be exploited by cancer cells. The high LAG-3 expression levels also seem to be related to the resistance to anti PD-1 ICIs and with a negative prognostic outcome [41]; a dual PD-1/LAG-3 blockade therapy could also represent a viable option. VISTA, on the other hand, can act as a ligand when expressed on the surface of APCs and, more importantly, as a receptor when presented on the surface of T cells (TILs included); the binding partner(s) of VISTA expressed on tumor cells are still not fully identified [42,43,44,45,46]. Quite interestingly, while the higher VISTA expression levels seem to be associated with the higher TILs levels, they also seem to be linked to a positive prognostic outcome [47]. Conversely, OX 40 and its ligand OX40L represent a positive IC. OX40 is expressed on the surface of T cells and TILs, while OX40L is expressed on the surface of APCs [48,49,50]. In a physiological setting the OX-40-OX40L pathway mediates the co-stimulatory signals for T-cells activation, survival, and activity [51,52]. TILs with high OX40 expression levels seem to correlate with a positive prognostic outcome [53]. Therefore, OX40 agonizing monoclonal antibodies could represent an interesting strategy aimed at boosting the patients’ immune system against cancer cells. Lastly, in the same vein, GITR (expressed on the surface of T cells and TILs) and its ligand GITRL (expressed on the surface of APCs) represent another positive IC and promote and regulate T-cells’ activation and activity. GITR agonists could provide interesting results in future clinical trials [54,55,56,57,58,59,60,61] (Table 1).

As previously stated, early clinical trials are starting to investigate ICIs targeting these new IC. While it is too early to draw definite conclusions, thanks to the above-mentioned preclinical data, the most promising strategy is represented by a dual targeting of both the traditional and new IC. The notable amount of ongoing trials underlines the great interest towards this new approach (Table 2).

## 3. Advanced NSCLC: Current Role of Immunotherapy

With reference to the above-mentioned EMA and/or FDA-approved ICIs, according to the most recent ASCO and ESMO guidelines, they are granted their strongest recommendations—alone or in combination with chemotherapy—in the first-line setting for naïve non-oncogene-addicted patients. Therefore, these regimens represent, by far, the most used treatments in a real-world scenario [62,63] (Table 3).

## 4. Prognostic Factors in Advanced NSCLC Patients Treated with ICIs: Current State of the Art and Future Possibilities

### 4.1. PD-L1

As the above-mentioned clinical trials and international recommendations show, tumor tissue PD-L1 is the only sub-optimal biomarker whose predictive role is taken into account when considering ICI treatment in advanced NSCLC patients; in fact, while the higher PD-1 expression rates are linked to better responses to ICI treatment, patients that do not express PD-L1 can still experience meaningful responses [76]. The same does not apply, however, to its prognostic value. In fact, while large systematic studies are lacking, the available data seem to show no correlation between the PD-L1 expression levels and the survival outcomes [77]. On the other hand, the different levels of soluble PD-L1 detected in the plasma of NSCLC patients receiving ICI seem to be associated with different responses. While the lower levels of soluble PD-L1 seem to be associated with a favorable prognostic value (longer PFS and OS-, higher ORR—Overall Response Rates), the higher levels seem to be linked to a worse prognosis. A soluble PD-L1 assessment could offer an easily repeatable test (as opposed to tumor tissue biopsy), while also allowing us to better follow the dynamic changes associated with the response to therapy or lack thereof; these assessments, however, are still in an experimental phase and far from standardization [78,79,80,81].

### 4.2. Tumor Mutational Burden

The role of Tumor Mutational Burden (TMB), meaning the amount of mutations per DNA-coding regions, has also been investigated in advanced NSCLC patients receiving immunotherapy, thanks to the preclinical evidence that higher mutations rates lead to the higher levels of neoantigens (“hot tumors”), and thus, to a higher activation of the immune system, rendering these tumors virtually more susceptible to ICI treatment [82,83,84]. Similarly to PD-L1, almost all the major clinical trials have been focused on assessing the TMB predictive value, yet without definite findings. In fact, while pembrolizumab monotherapy in patients with ≥175 mutations per exome proved to be associated with a positive predictive value in the KEYNOTE-010 and 042 trials, this was not the case in the KEYNOTE-021 trial [85]. The combination of pembrolizumab + chemotherapy was similarly not associated with a positive predictive value in the KEYNOTE-189 and 407 trials [86].

In the same vein, the biggest trial investigating the TMB role in this setting was the CheckMate 227 study, assessing the nivolumab + ipilimumab association in 1189 naïve NSCLC patients with a TMB ≥ 10 mutations per megabase. When compared to standard chemotherapy, this association managed to provide superior results in terms of ORR, PFS, and OS, seemingly establishing a high TMB as a predictive factor in this setting. However, further data demonstrated that this benefit is also associated with a TMB < 10 mutations per megabase, albeit with a lower magnitude (median OS: 23.03 months vs. 16.72 months and 16.20 months vs. 12.42, respectively), thus possibly redefining TMB as a prognostic factor. Once again, the lack of standardization in terms of the cut-off represents a key problem [87,88].

### 4.3. Neutrophil-to-Lymphocytes Ratio

The preclinical data show that while lymphocytes (and particularly CD8+ ones) are the main agents involved in cancer cells killing both directly and through cytokines (e.g., IFN-γ, IL-2, and IL-12), neutrophiles foster a pro-cancer inflammatory microenvironment that inhibits lymphocytes activity via the production of a palette of cytokines (e.g., IL-10, TNF-α, and VEGF) [89,90,91]. In this sense, an elevated (albeit with different cut-offs, ranging from NLR ≥ 3 to NLR ≥ 5) Neutrophil-to-Lymphocyte Ratio (NLR) has been associated with a poor prognostic value in NSCLC patients receiving chemotherapy, targeted therapy, and immunotherapy. In fact, NSCLC patients with a high NLR receiving ICI consistently experience shorter PFS and OS, as the data coming both from single experiences and meta-analyses show [92,93,94,95,96]. For example, in a recent and large Italian retrospective trial, 252 advanced NSCLC patients receiving pembrolizumab in the first line setting were assessed, employing a cut-off of NLR ≥ 4.8. As a result, patients with a high NLR presented decisively lower OS results when compared to their low NLR counterparts: mOS 7.6 months vs. 34.8 months; interestingly, patients with a very high NLR (>10) were associated with a dismal prognosis: mOS: 3.8 months [97].

### 4.4. Tumor-Infiltrating Lymphocytes

Building on the same preclinical evidence, tumor-infiltrating lymphocytes (TILs) could represent another valid prognostic factor in advanced NSCLC patients receiving ICIs [89,90,91,98].

This hypothesis has been explored in a recent study in which 26 advanced NSCLC patients receiving immunotherapy were enrolled. Patients whose tumor tissue presented more than 886 CD8+ lymphocyte/mm^2^ were considered as high expressors, while patients with less than 886 CD8+ lymphocyte/mm^2^ were considered low expressors. Low expressors presented lower response rates to ICIs when compared to the high expressors (16.7%, vs. 60%); moreover, patients whose tumor tissue presented CD8+/CD4+ ratios lower than two showed lower response rates than patients whose tumor tissue presented a >2 ratio RR (13.3% vs. 43 to 50%) [99]. While interesting, these factors need further validation and standardization in larger trials.

### 4.5. Combined Scores

Apart from the NLR, other agents responsible for the pro-cancer inflammatory microenvironment are represented by high LDH and C-reactive protein levels; these markers are also evaluated in different scores in combination with other factors that are classically linked to poor prognosis in solid tumors, NSCLC included, like hypoalbuminemia and low platelet count [100,101,102,103,104,105]. As of today, several different combined scores have provided interesting results in NSCLC patients receiving ICIs. The Glasgow Prognostic Score (GPS) combines c-reactive protein levels >10 mg/L and hypoalbuminemia (<35 g/L), defining three different scores: 0: CRP < 10 mg/L and albumin > 35 g/L; 1: CRP ≥ 10 mg/L or albumin < 35 g/L; or 2: CRP > 10 mg/L and albumin < 35 g/L; higher GPS are associated with lower PFS and OS results in patients receiving ICIs [106,107]. Similarly, the Lung Immune Prognostic Index (LIPI) takes into account NLR (>3 as cutoff) and LDH (>upper limit of normal as cutoff), defining three scores: good (0 factor), intermediate (1 factor), and poor (2 factors); in several large studies, higher scores represent a poor prognostic factor [108,109,110].

### 4.6. Gut Microbiota

An ever-growing set of preclinical data underlines the role of gut microbiota in regulating adaptive and innate immunity, and thus, in partially positively regulating immune responses against cancer cells [111,112,113]. However, as of today, neither the precise mechanisms of action, the specific bacteria, nor the possible positive prognostic value are well understood. Amidst conflicting data and theories, however, some species seem to be more prevalent in patients benefitting from immunotherapy (Bifidobacterium breve/adolescentis/longum) and they seem to exert this favorable effect by improving the production of anti-cancer cytokines like IFN-γ and possibly by cross-reactions between bacteria antigens and tumor ones [114,115,116,117].

### 4.7. Concomitant Medications

Several types of drugs have been postulated to negatively interfere with the survival outcomes in solid tumors—NSCLC included—treated with ICIs, but Proton Pump Inhibitors (PPIs), corticosteroids (CCS), and antibiotics (ATBs) are the classes that present the most solid evidence and that thus seem to represent some negative prognostic factors. While the CCS’s immune-suppressive role is well documented and understood and suffices to explain their negative impact on ICIs’ activity, PPIs’ and ATBs’ interferences with immunotherapy effects are still being investigated, but their modulating effects on the gut microbiome seem to play a crucial role [118,119,120,121,122,123,124]. With specific respect to advanced NSCLC, a very recent and large retrospective trial enrolling 950 advanced naïve NSCLC receiving pembrolizumab further confirmed these findings, linking shorter OS to CCS’s, ATBs’, and PPIs’ treatment [125].

### 4.8. High Body Mass Index

Obesity has been shown to be associated with a chronic state of inflammation and this inflammation affects the host’s immune response, influencing ICIs’ efficacy in solid tumors—NSCLC included. In particular, contrarily to most of the factors listed up to this point, obesity (defined as patients’ BMI ≥ 30) seems to act as a positive prognostic factor (longer PFS and OS, higher ORR) in cancer patients because it boosts the host’s immune system, and thus, it ultimately enhances ICIs activity. While the exact mechanisms are still not completely understood, leptin seems to play a key role. In fact, the enhanced leptin signaling found in obese hosts leads to an increased number of exhausted T lymphocytes (exhaustion which could be at least partially mediated via immune checkpoints); however, quite paradoxically, these exhausted T lymphocytes seem to be more easily re-activated and boosted by ICI treatment [126,127,128,129,130,131,132,133,134,135]. A recent and extensive international retrospective study confirmed these findings in a large real-world cohort of 962 naïve advanced NSCLC patients receiving pembrolizumab. In fact, obese patients (BMI ≥ 30) were found to present longer PFS and OS and higher ORR rates [136]. Very interestingly, this favorable prognostic role does not seem to be associated with overweight patients (BMI ≥ 25). In this vein, a recent retrospective Japanese trial assessed 324 advanced NSCLC patients treated with PD-1 inhibitors, dividing them in a non-overweight group (BMI < 25; 86.1%, 279 patients) and in an overweight one (13.9%, 45 patients). As a result, no significant differences were found between the two groups in terms of ORR, PFS, and OS [137].

### 4.9. Weight Loss

While weight loss and cancer cachexia (mediated by cancer-linked pro-inflammatory cytokines) are well-established negative prognostic factors in advanced NSCLC patients receiving chemotherapy [138,139,140,141], they also seem to represent negative prognostic factors in advanced NSCLC patients receiving ICI. In fact, several prospective and retrospective international trials have consistently linked the lower ORR, PFS, and OS rates to weight loss and cachexia in ICI-receiving patients. While the exact mechanisms behind this relation are not fully understood, the pro-inflammatory microenvironment leading to cachexia seems to play a central role in inhibiting the host’s immune system; in particular, these pro-inflammatory cytokines seem to particularly affect the NLR and the gut microbiome composition [142,143,144,145,146,147] (Table 4).

## 5. Conclusions

Immunotherapy has represented an incredible revolution in the field of advanced NSCLC treatment; however, as of today (and as the most recent guidelines show), we lack effective biomarkers to both more effectively select and manage patients who could benefit the most from immunotherapy. As the above-mentioned data shows, several different factors have been investigated, but easily assessable ones like NLR, BMI, and the different combined scores not only present the most robust data, but also could allow us to dynamically and periodically re-evaluate patients. The data derived from these biomarkers are unlikely to lead to new strategies and to change our clinical practice; however, they could represent a valid addition to our armamentarium for a more efficient patient selection. In this sense, the comprehensive combined scores or panels could represent the most viable option for optimizing the vast amount of different factors while taking into account their complex interplay. The biggest problem regarding these factors, however, is represented by the current lack of validation and standardization, which should be addressed with larger and prospective trials [148,149].

## Figures and Tables

**Table 1 cancers-15-04684-t001:** Currently clinically targetable Immune Checkpoint and future candidates.

IC	Binding Partners Expressed Respectively on	Role	Targetability
PD-1—PD-L1	T cells—tumor cells	Negative IC	Currently targetable
CTLA-4—B7	T cells—tumor cells	Negative IC	Currently targetable
TIM-3—Galectin-9	T cells—tumor cells	Negative IC	Still not targetable
LAG3—class II MHC/galectin-3/LSECtin	T cells—tumor cells	Negative IC	Still not targetable
VISTA—Still unknown	T cells	Negative IC	Still not targetable
OX40—OX40L	T cells—APCs	Positive IC	Still not targetable
GITR—GITRL	T cells—APCs	Positive IC	Still not targetable

**Table 2 cancers-15-04684-t002:** Clinical trial investigating ICIs targeting new ICs.

IC	ICI	Design	Identifier
TIM-3	LY3415244 (anti TIM-3 and PD-L1)	LY3415244 in advanced or metastatic solid neoplasms	NCT03752177
TIM-3	LY3321367 (anti TIM-3)	LY3321367 ± LY3300054 (anti PD-L1 ICI) in advanced or metastatic solid neoplasms	NCT03099109
TIM-3	Sym023 (anti TIM-3)	Sym023 in advanced or metastatic solid neoplasms or lymphomas	NCT03489343
TIM-3	TSR-022 (anti TIM-3)	TSR-022 + TSR-042 (anti PD-L1 ICI) + carboplatin + pemetrexed/nab-paclitaxel in advanced or metastatic solid neoplasms	NCT03307785
TIM-3	Sym023 (anti TIM-3)	Sym023 + Sym021 (anti PD-1 ICI) in advanced or metastatic solid neoplasms or lymphomas	NCT03311412
TIM-3	TSR-022 (anti TIM-3)	TSR-022 ± TSR-042 (anti PD-L1 ICI) / TSR-022 ± TSR-033 (anti LAG3 ICI) in advanced or metastatic solid neoplasms	NCT02817633
TIM-3	RO7121661 (anti TIM-3 and PD-1)	RO7121661 in advanced or metastatic solid neoplasms	NCT03708328
TIM-3	INCAGN02390 (anti TIM-3)	INCAGN02390 in advanced or metastatic solid neoplasms	NCT03652077
TIM-3	MBG453 (anti TIM-3)	MBG453 ± PDR001 (anti PD-1 ICI) in advanced or metastatic solid neoplasms	NCT02608268
TIM-3	BGB-A425 (anti TIM-3)	BGB-A425 ± tislelizumab (anti PD-1 ICI) in advanced or metastatic solid neoplasms	NCT03744468
LAG 3	Sym022 (anti LAG 3)	Sym022 in advanced or metastatic solid neoplasms or lymphomas	NCT03489369
LAG 3	LAG525 (anti LAG 3)	LAG525 ± PDR001 (anti PD-1 ICI) in advanced or metastatic solid neoplasms	NCT02460224
LAG 3	IMP321 (anti LAG 3)	IMP321 in advanced or metastatic solid neoplasms	NCT03252938
LAG 3	Relatlimab (anti LAG 3)	Relatlimab ± nivolumab in advanced or metastatic solid neoplasms	NCT02966548
LAG 3	XmAb^®^22841 (anti LAG 3 and CTLA-4)	XmAb^®^22841 ± pembrolizumab in advanced or metastatic solid neoplasms	NCT03849469
LAG 3	Relatlimab (anti LAG 3)	Relatlimab ± nivolumab in advanced or metastatic solid neoplasms	NCT01968109
LAG 3	MGD013 (anti LAG 3)	MGD013 in advanced or metastatic solid neoplasms or lymphomas	NCT03219268
VISTA	CA-170 (anti VISTA and PD-L1 and PD-2)	CA-170 in advanced or metastatic solid neoplasms or lymphomas	NCT02812875
OX40	MEDI0562 (anti OX-40)	MEDI0562 + durvalumab / MEDI0562 + tremelimumab in advanced or metastatic solid neoplasms	NCT02705482
OX40	MOXR0916 (anti OX-40)	MOXR0916 + atezolizumab in advanced or metastatic solid neoplasms	NCT02410512
OX40	INCAGN01949 (anti OX-40)	INCAGN01949 + nivolumab / INCAGN01949 + ipilimumab / INCAGN01949 + nivolumab + ipilimumab in advanced or metastatic solid neoplasms	NCT03241173
OX40	PF-04518600 (anti OX-40)	PF-04518600 ± 05082566 (4-1BB ICI) in advanced or metastatic solid neoplasms	NCT02315066
OX40	ATOR-1015 (anti OX-40)	ATOR-1015 in advanced or metastatic solid neoplasms	NCT03782467
OX40	SL-279252 (anti OX-40)	SL-279252 in advanced or metastatic solid neoplasms or lymphomas	NCT03894618
OX40	ATOR-1015 (anti OX-40)	ATOR-1015 in advanced or metastatic solid neoplasms	NCT03782467
OX40	SL-279252 (anti OX-40)	SL-279252 in advanced or metastatic solid neoplasms or lymphomas	NCT03894618
GITR	INCAGN01876 (anti GITR)	INCAGN01876 in advanced or metastatic solid neoplasms	NCT02697591
GITR	TRX518 (anti GITR)	TRX518 in advanced or metastatic solid neoplasms	NCT01239134
GITR	GWN323 (anti GITR)	GWN323 ± PDR001 (anti PD-1 ICI) in advanced or metastatic solid neoplasms or lymphomas	NCT02740270
GITR	OMP-336B11 (anti GITR)	OMP-336B11 in advanced or metastatic solid neoplasms	NCT03295942

**Table 3 cancers-15-04684-t003:** Current ESMO and/or ASCO-recommended ICIs or ICI-containing combinations for the treatment of naïve advanced NSCLC without driver mutations.

ICI or ICI-Containing Combination	Subset of Patients	Regulatory Approval	Pivotal Trial	Updated Survival Data (mPFS/OS) *
Pembrolizumab	Squamous and nonsquamous with PD-L1 TPS ≥ 50%	ASCO-recommended ESMO-recommended	KEYNOTE-024 [64]	PFS: 7.7 months OS: 26.3 months
Atezolizumab	Squamous and nonsquamous with PD-L1 TPS ≥ 50%	ASCO-recommended ESMO-recommended	IMpower110 [65]	PFS: 8.2 months OS: 18.9 months
Cemiplimab	Squamous and nonsquamous with PD-L1 TPS ≥ 50%	ASCO-recommended ESMO-recommended	Empower-Lung 1 [66]	PFS: 6.3 months OS: 23.4 months
Pembrolizumab	Squamous and nonsquamous with any PD-L1 TPS	ASCO-recommended	KEYNOTE-042 [67]	PFS: 5.6 months OS: 16.4 months
Nivolumab + Ipilimumab + short course of platinum-based histology-specific doublet chemotherapy followed by Nivolumab + Ipilimumab	Squamous and nonsquamous with any PD-L1 TPS	ASCO-recommended ESMO-recommended	CheckMate 9LA [68]	PFS: 6.4 months OS: 15.8 months
Cemiplimab + platinum-based histology-specific doublet chemotherapy followed by Cemiplimab (+Pemetrexed if nonsquamous)	Squamous and nonsquamous with any PD-L1 TPS	ESMO-recommended	Empower-Lung 3 [69]	PFS: 8.2 months OS: 21.1 months
Durvalumab + Tremelimumab + platinum-based histology-specific doublet chemotherapy followed by Durvalumab + one additional Tremelimumab dose (+Pemetrexed if nonsquamous)	Squamous and nonsquamous with any PD-L1 TPS	ESMO-recommended	POSEIDON [70]	PFS: 6.2 months OS: 14.0 months
Nivolumab + Ipilimumab	Squamous and nonsquamous with a PD-L1 TPS ≥ 1%	ESMO-recommended	CheckMate 227 [71]	PFS: 5.1 months OS: 17.1 months
Pembrolizumab + Platinum + Pemetrexed followed by Pembrolizumab + Pemetrexed	Nonsquamous with any PD-L1 TPS	ASCO-recommended ESMO-recommended	KEYNOTE-189 [72]	PFS: 9.0 months OS: 22.0 months
Atezolizumab + Carboplatin + Nab-Paclitaxel followed by Atezolizumab	Nonsquamous with any PD-L1 TPS	ASCO-recommended ESMO-recommended	IMpower130 [73]	PFS: 7.0 months OS: 18.6 months
Atezolizumab + Carboplatin + Paclitaxel + Bevacizumab followed by Atezolizumab + Bevacizumab	Nonsquamous with any PD-L1 TPS	ASCO-recommended ESMO-recommended	IMpower150 [74]	PFS: 8.4 months OS: 19.5 months
Pembrolizumab + Carboplatin + (Nab)Paclitaxel followed by Pembrolizumab	Squamous with any PD-L1 TPS	ASCO-recommended ESMO-recommended	KEYNOTE-407 [75]	PFS: 8.0 months OS: 17.2 months

* mPFS: Median Progression Free Survival; mOS: Median Overall Survival.

**Table 4 cancers-15-04684-t004:** Most promising prognostic factors under investigation in advanced NSCLC patients treated with ICIs.

Factor	Prognostic Value
High soluble PD-L1 levels	Negative
High Tumor Mutational Burden	Positive
High Neutrophil-to-lymphocytes ratio	Negative
High levels of Tumor-infiltrating lymphocytes	Positive
High Glasgow Prognostic Score	Negative
High Lung Immune Prognostic Index	Negative
Gut microbiota	Debated
Corticosteroids, antibiotics, and PPI use	Negative
High Body Mass Index	Positive (BMI ≥ 30)
Weight loss and cachexia	Negative

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
