# Peer review of "Prognostic Factors in Advanced Non-Small Cell Lung Cancer Patients Treated with Immunotherapy"

_cancers, 2023, doi:10.3390/cancers15194684_

Round 1
Reviewer 1 Report
1. In the first, paragraph, there is no need to state that the second most common cancer in males and females combined.
2. In line 92, it would be appropriate to say that PDL 1 is a "sub-optimal" predictive marker in place of "albeit flawed..."
3. In the section discussing the relationship between BMI and outcomes in NSCLC patients treated with immune check point inhibitors, it would be good to include the following reference: Tateishi A et al. Resp. Invest 60; 2022, 234 - 240. These investigators found that being overweight defined as BMI > 25 was not associated with superior survival in NSCLC patients treated with pembrolizumab.
4. Weight loss has also been shown to be associated with inferior survival in NSCLC patients treated with immune checkpoint inhibitors. It would be appropriate to include the following references and discussion of their results - Lee CS Future Medicine 2020; Miyawaki T J Thorac Oncol 2020; and Jo H Cancer Immunology Immunotherapy 2022.
5. In your conclusion, you appropriately state that additional study of these prognostic factors is needed. Do yo suspect that these studies could identify a panel of "predictive markers" or could lead to novel therapeutic strategies.
Author Response
First of all thank you for your constructive criticism.
1) Revised accordingly
2) Revised accordingly
3) Reference included and discussed
4) Reference included and discussed
5) Conclusion revised according to this suggestion
Reviewer 2 Report
In the review manuscript entitled “prognostic factors in advanced non-small cell lung cancer patients treated with immunotherapy”, the authors summarized the prognostic factors which can predict the efficacy of ICIs in NSCLC. It was well-written. However, there were lots of review papers covering this topic. Thus, it lacks novelty.
Author Response
First of all thank you for your constructive criticism.
1) In this sense, we added a section discussing novel immune checkpoints
Reviewer 3 Report
Major Issues:
1. Authors presenting findings did not challenge current thinking.
2. The inhibitors of other immune checkpoints(TIM-3,LAG-3 and so on) are also very important in clinical research. The author should describe it.
Minor Issues
3. Please double check the English expression of naïve(Table 1).
4. The abbreviations in the manuscript should be described when they first appear.(CTLA-4).
Tables, language and manuscript structure were not clear enough.
Author Response
First of all thank you for your constructive criticism
1-2) We added a new section discussing novel immune checkpoints and revised the conclusion
3) Double checked
4) Revised accordingly
Round 2
Reviewer 2 Report
The novelty of this review paper is not significantly improved.